# The Influence of New Silicate Cement Mineral Trioxide Aggregate (MTA Repair HP) on Metalloproteinase MMP-2 and MMP-9 Expression in Cultured THP-1 Macrophages

**DOI:** 10.3390/ijms22010295

**Published:** 2020-12-30

**Authors:** Katarzyna Barczak, Mirona Palczewska-Komsa, Mariusz Lipski, Dariusz Chlubek, Jadwiga Buczkowska-Radlińska, Irena Baranowska-Bosiacka

**Affiliations:** 1Department of Conservative Dentistry and Endodontics, Pomeranian Medical University in Szczecin, Powstańców Wlkp. 72, 70-111 Szczecin, Poland; kasiabarczak@vp.pl (K.B.); mpalczewskakomsa@op.pl (M.P.-K.); zstzach@pum.edu.pl (J.B.-R.); 2Department of Preclinical Conservative Dentistry and Preclinical Endodontics, Pomeranian Medical Univesity in Szczecin, Powstańców Wlkp. 72, 70-111 Szczecin, Poland; lipam@pum.edu.pl; 3Department of Biochemistry and Medical Chemistry, Pomeranian Medical University in Szczecin, Powstańców Wlkp. 72, 70-111 Szczecin, Poland; dchlubek@pum.edu.pl

**Keywords:** mineral trioxide aggregate (MTA Repair HP), bioactive calcium-silicate cement, THP-1 monocytes, macrophages, metalloproteinase 2 (MMP-2), metalloproteinase 9 (MMP-9), inflammatory reaction

## Abstract

The aim of the present study was to investigate the new silicate cement mineral trioxide aggregate (MTA Repair HP) with respect to its effect on the inflammation process involving the tooth and periodontal tissues. The composition of MTA Repair HP was supplemented with plasticizer agents which can have a negative effect on the modulation of tooth inflammation. The silicate-based material in question is widely used in regeneration of the pulp-dentin complex, treatment of perforations of various locations in the tooth, as well as in surgical treatment of the complications of periapical tissue. The improved bioceramic restorative cement can affect the expression of metalloproteinases MMP-2 and MMP-9 in monocytes/macrophages involved in modulation of inflammation and regenerative processes of the tooth and periodontal tissues. The novel aspect of the present study lies in the application of the model of THP-1 monocyte/macrophage and applying the biomaterial in direct contact with the cells. Hence, it provides a representation of clinical conditions with respect to regenerative pulp and periodontal treatment with the use of MTA Repair HP. A lack of macrophage activation (as measured with flow cytometry) was found. Moreover, the study identified a lack of expression stimulation of the studied metalloproteinases (with the use of Western blotting and fluorescent microscopy). Similarly, no increase in MMP-2 and MMP-9 concentration was found (measured by ELISA method) in vitro when incubated with MTA Repair HP. Based on the results it can be concluded that new MTA Repair HP does not increase the inflammatory response in monocytes/macrophages associated with the activity of the described enzymes. It can also be speculated that they do not affect the process of dentin regeneration in which MMP-2 and MMP-9 play significant roles.

## 1. Introduction

There is an exponential increase in the clinical uses of bioceramics due to a wide range of applicability of the materials in restorative dentistry and endodontics. The introduction of mineral trioxide aggregate (MTA) allowed further enhanced application and significant improvement of properties with the purpose of achieving the maximum benefits. Numerous reports and studies confirm that with respect to various clinical procedures, MTA is considered as the golden standard material [1,2]. With respect to its biocompatibility, bioactivity and hydrophilicity, as well as radiodensity, sealing ability and low solubility, MTA shows several desirable properties. High biocompatibility, as was reported in scientific studies, encourages optimal healing response [3]. Contrary to other dental materials, MTA is known to set in a moist environment. When in contact with moisture, the chief component of MTA, i.e., calcium oxide, converts into calcium hydroxide, well-known to many clinicians. This results in a high pH microenvironment which was shown to have desirable antibacterial effects. However, unlike calcium hydroxide, this material shows very low solubility and is additionally known to maintain its physical integrity after placement [3]. MTA can be used both in surgical as well as non–surgical applications, e.g., direct pulp capping, as a temporary filling material, in perforation repairs in roots or furcations, for apexification procedure and root end fillings (see Appendix A). The antibacterial and antifungal properties of MTA can be attributed to its pH. This, in turn, shows an inhibitory effect on the growth of microorganism and results in disinfection of dentin [4,5].

Treatment of deep caries requires the use of materials with biochemical and physical properties which are as close as possible to the respective parameters of dentin. Although new materials and techniques continue to be introduced to the market, side effects are still reported. The example could be an immune response which can lead to aggravation of inflammation and, consequently, to premature extraction of the tooth [6,7,8,9]. In a healthy tooth pulp there are cells which take part in immunological reactions, i.e., macrophages, lymphocytes, plasma cells and mast cells. Macrophages originate from monocytes circulating in blood and spread into the pulp. They are present in close vicinity to the walls of the blood vessels and capillaries. In physiological conditions, macrophages in the pulp are latent. However, under pathological conditions, they show mobility and migrate to the inflammation site [6,7,10,11]. The use of biomaterials can cause cytokines and chemokines to be released as well as result in the activation of the complement system, thus exacerbating the inflammation [6,7,8,9]. Monocytes/macrophages are among the first cells which come in contact with the material used for following its placing on an exposed or partially damaged pulp [12]. Development of pulp inflammation is closely connected with the activation of macrophages [13,14,15,16]. Release of cytokines and growth factors by macrophages initiates both the reparative as well as destructive processes [17,18,19]. As early as on the initial contact of the covering material with the pulp, and intense release of proinflammatory cytokines is observed—IL-1β, IL-6, TNF-α and chemokine MCP-1 and MIP-1α (chemoattractant protein of monocytes-1 and inflammatory protein of macrophages-1α [17,19,20]. It is established that the reconstruction of the dental matrix depends on several factors such as matrix metalloproteinases (MMP), enzymes involved in remodeling of the bone, cells synthesized by osteoclasts as well as other cells e.g., fibroblasts and macrophages [21,22].

The main components of extracellular matrix (ECM) serve as the main substrates for metalloproteinases. The structural changes in the matrix results from the MMP-induced catalysis. Other functions of MMPs have been identified, just to mention the regulation of cell activity or participation in inflammatory reactions. Many peptides released during partial ECM proteolysis may affect cell activity. In addition to ECM digestion, metalloproteinases induce the catalysis of other biologically active substances, e.g., growth factors or cell adhesion molecules [22]. The crucial regulatory role of MMPs in terms of cell differentiation and migration processes, growth factors, angiogenesis as well as development of inflammation has been confirmed [23,24,25,26].

The aim of the present study was to investigate the MTA Repair HP formula with respect to its effect on the inflammation process involving the tooth and periodontal tissues. The composition of MTA Repair HP was supplemented with plasticizer agents which can have a negative effect on the modulation of tooth inflammation. The silicate-based material in question is widely used in regeneration of the pulp-dentin complex, treatment of perforations of various locations in the tooth, as well as in surgical treatment of the complications of periapical tissue. The improved bioceramic restorative cement can affect the expression of metalloproteinases MMP-2 and MMP-9 in monocytes/macrophages involved in modulation of inflammation and regenerative processes of the tooth and periodontal tissues. The novel aspect of the present study lies in the application of the model of monocyte/macrophage THP-1 with biomaterial applied in direct contact with the cells. Therefore, it provides a representation of clinical conditions with respect to regenerative pulp as well as periodontal treatment using MTA Repair HP.

## 2. Results

### 2.1. MTA Repair HP-Induced Activation of THP-1 Monocytes

The results of the analysis conducted using flow cytometry indicated that monocytes incubated in the presence of MTA Repair HP did not show a statistically significant increase in the expression of CD14 receptor (a marker for monocyte differentiation), (*p* = 0.65), whereas the expression of CD68 (a marker for macrophage differentiation) increased significantly following 48 h in cells treated with PMA. Activation of THP-1 monocytes by MTA Repair HP present in the culture was not found (Figure 1, Table 1).

### 2.2. Expression of MMP-2 in Monocytes

Incubation time of monocyte culture in the control conditions showed a marked effect on expression of protein MMP-2. Enzyme expression increased more than 2.5 times (*p* ≤ 0.001) in monocytes incubated for 48 h in the control conditions (control 48) vs. monocytes cultured in the control conditions for 24 h (control 24). MMP-2 expression was also markedly higher in monocytes incubated in the presence of MTA Repair HP 24 h groups (MTA24) vs. monocytes cultured in the control conditions (control 24)—the increase was approximately 120%, *p* ≤ 0.001. However, comparison of the control 48 and MTA Repair HP 48 h groups (MTA48) showed that enzyme expression was significantly lower (by approx. 25%, *p* ≤ 0.001). Monocytes incubated in the presence of MTA Repair HP for 24 and 48 h (MTA24 vs. MTA48) did not show statistically significant changes in expression of the analyzed enzyme. No visible differences in enzyme protein (MTA24 vs. MTA48 group) were observed in the image from the fluorescence microscopy, (Figure 2 and Figure 3, see also Appendix A, *1S. MMP-2 and MMP9 activity in macrophages measurement by zymography method,*
Appendix A).

### 2.3. Expression of MMP-2 in Macrophages

Incubation time of macrophage culture in the control conditions showed a marked effect on the expression of MMP-2 protein, which was higher by approx. 50% (*p* ≤ 0.001) in control 48 macrophage vs. control 24. No significant changes in the expression of enzyme between control 24 vs. MTA24, control 48 vs. MTA48 were found (Figure 4 and Figure 5, see also Appendix A).

### 2.4. Expression of MMP-9 in Monocytes

Incubation time of monocyte culture in the control conditions showed a significant effect on the expression of MMP-9 protein. Enzyme expression increased by more than 50% (*p* ≤ 0.001) in control 48 vs. control 24. MMP-9 expression was markedly lower in monocytes incubated in the presence of MTA24 vs. control 24 (an increase by approx. 50%, *p* ≤ 0.001). The comparison of the control 48 and MTA 48 groups shows that enzyme expression was significantly lower (by approx. 45%, *p* ≤ 0.001). In monocytes incubated in the presence of MTA Repair HP for 24 and 48 h (MTA24 vs. MTA48), there were no statistically significant changes in the expression of the enzyme under analysis (Figure 6 and Figure 7, see also Appendix A).

### 2.5. Expression of MMP-9 in Macrophages

Incubation time of macrophage culture in the control conditions showed a marked effect on MMP-9 protein expression, which was higher by approx. 40% (*p* ≤ 0.001) in macrophages control 48 vs. control 24. There were no significant changes in the expression of enzyme between control 24 vs. MTA24 and control 48 vs. MTA48 (Figure 8 and Figure 9, see also Appendix A).

## 3. Discussion

The non-mineralized pulp serves numerous functions, including induction and dentin formation. It also has nutritional, protective and sensory functions for the structures of the tooth. Both accidental mechanical exposing of the pulp as well as caries exposure may lead to irreversible damage to the tissue, unless properly treated [27]. The biological treatment of the pulp is based on the use of biocompatible materials which allow continued proper functioning of the exposed pulp providing the bioindicator effect and repair stimulation by the formation of new mineralized tissue. The proper structure and thickness of the newly formed dentine barrier (dentin bridge) is not always positively correlated with the clinical success. From the clinical perspective, the most important effect are the long-term pulp vitality and absence of symptoms [27]. It must be observed that the covering material is in constant contact with the pulp. Therefore, the toxicity/safety of the covering material with respect to pulp is of crucial importance [28].

Pulp damage due to caries or mechanical damage is connected with the destruction of the tissue integrity which results in excessive synthesis of proinflammatory cytokines and prostaglandins which can initiate the development of inflammation and, in turn, lead to further damage of the tooth tissue [27,28]. Owing to this, it is vital that the materials used for pulp repair do not cause additional inflammation and, at the same time, do not interfere with the regeneration process. The matrix metalloproteinases MMP-2 and MMP-9 tested in the present study also contribute to inflammatory processes in tissues by interacting with chemokines. MMP-2 removes the N-terminal amino group of chemotactic tetrapeptide of monocyte-chemotactic protein 3 (MCP3). Such modified MCP3 becomes a chemokine receptor antagonist, and results in the extinction of the inflammatory reaction [24]. This mechanism is known to determine the chemokine-mediated contribution of MMP-2 to the inflammatory reactions. As a result of the activation of MMP-9 in the context of inflammatory reactions, a dissimilar effect is reported. Since MMP-9 removes the six amino acids from the N-terminus of interleukin-8 (IL-8), the shortened form of IL-8 becomes a much stronger stimulus and causes an increased inflow of neutrophils [25]. It can be inferred that the increased expression of this enzyme through activation of inflammatory interleukins is a characteristic of the initial stage of inflammation [25]. The extinction of the inflammatory reaction can be linked to the increased MMP-2 synthesis [29]. In the present study, the lack of significant changes in the expression of MMP-9 and MMP-2 enzymes in THP-1 macrophages incubated in the presence of the examined MTA indicates the lack of additional induction of expression of these enzymes by the material—which can indicates its biocompatibility. The introduction of the MTA preparation to dentistry in the 1990s was a milestone in increasing the effective treatment of iatrogenic complications arising during root canal treatment. MTA provides an excellent solution due to its high biocompatibility, water-binding capacity, good sealing ability, bioactivity and other desirable properties for clinical applications. Despite its advantages, MTA’s disadvantages are discoloration, long setting time, low flow ability and difficult handling. This has led to the need to improve physical properties and develop an improved formula to overcome these drawbacks. In response to this, other preparations were developed, such as “white MTA” or MTA Angelus HP [26].

The minimum toxicity without causing a significant inflammatory reaction is a basic principle of biocompatible materials. Biocompatibility is one of the most beneficial properties of the materials used in endodontic treatment. It is of immense importance due to prolonged contact of the material used for pulp capping. Pulp vitality is essential for proper functioning of teeth, and biocompatible materials play a crucial role in repair regrowth and covering of the pulp exposed due to injury with a new mineralized tissue, thus allowing regeneration of the damaged teeth [27].

MTA is the preferred material to be used for direct pulp capping. It was found that the use of MTA as a direct pulp capping agent, it shows a far better result with respect to complete formation of dentin bridge and decreasing inflammation, in comparison with Dycal (calcium hydroxide) [30]. Other studies on MTA show that it is characterized by the ability to bind in the presence of water, good tightness and bioactivity [25]. Hydrophilic properties of MTA are of significant importance due to the fact that during endodontic properties, there is always some moisture present. This feature of the product allows its use in root canal with exudate occurring when treating teeth with necrotic pulp or inflammation of periapical tissues [25,31]. It was shown that MTA, as well as its derivatives, when in contact with a soft tissue, breaks down to calcium hydroxide thus giving alkaline pH. High pH of such materials may bring several biological advantages. Firstly, it can induce formation of hard tissues such as foci of calcified tissue obliterating the apical foramen. Secondly, it alters pH of the environment (in this case, dentin) to more alkaline which may interfere with the function of osteoclasts and favor alkalinization in the adjoining tissues thus facilitating healing.

Other studies also show that calcium silicate based cements, such as MTA, result in decreased inflammation as compared with resin based root canal fillers [31]. In view of this, MTA can be considered the golden standard and as a root-end filling material after periapical surgery [11,32]. MTA Repair HP demonstrates proper cytological compatibility with human pulp stem cells (hDPSC). Previously, it was demonstrated that MTA Repair HP can promote a biological response in hDPSC in terms of cell proliferation, morphology, migration and binding, while the material is still cytocompatible [33].

Both systemic as well as local factors may contribute to modulation of inflammation and biocompatibility of MTA. The study by Cosme-Silva et al., 2019 shows that arterial hypertension may increase tissue inflammation and, consequently, reduce the biomineralization ability of the calcium silicate based cements [34].

The host’s response depends on the properties of biomaterials, and the presence of small amount of metal oxides (aluminum, arsenic, beryllium, cadmium, chromium and iron) in MTA may constitute a potential flaw [34,35]. Moreover, the presence of bismuth oxide as a contrast medium for RTG in MM-MTA does not favor cell proliferation in cell culture and can increase their cytotoxicity [34,36]. It was found that calcium tungsten present as radiopacific agent in MTA Repair HP contributed to an increased calcium release in the initial stage, thus promoting increased biomineralization and improved resistance of the said material. MTA Repair HP cytotoxicity test shows an increased cell viability with biocompatibility and biomineralization comparable to that of white MTA Angelus [34].

The study by Talabani et al., 2020 indicates that iatrogenic pulp damage treated with calcium silicate based cement is generally free of severe degenerative reactions of tissue a month after treatment completion [29]. Shi et al., 2014 compared cytotoxic activity of iRoot BP Plus and MTA ProRooton human gingival fibroblasts in medium in vitro and evaluated the cells vitality. The results obtained in the aforementioned study demonstrate that cell vitality at different concentrations of iRoot BP Plus and MTA amounted to from 77.3% to 113.8% with no cytotoxic effects recorded in either of the materials [37].

The fact that MTA tested in the present study did not show an increase in the synthesis of the MMP-2 and MMP-9 enzyme is yet another proof of their safety. It additionally confirms the results of our previous in vitro studies showing that Biodentine^TM^, being a representative of a silicate cement group, does not increase the expression of COX1 and COX2 enzymes in THP-1 monocytes/macrophages and does not initiate inflammation via the cyclooxygenase pathway (COXs) [38].

In the present study, the presence of the MTA Repair HP in the cultures of THP-1 monocytes/macrophages did not increase the expression of MMP-2 or MMP-9. Based on the results of our previous studies as well as the present study, it can be concluded that MTA Repair HP does not increase the inflammatory response in monocytes/macrophages associated with the activity of the described enzymes. It can also be speculated that they do not affect the process of dentin regeneration in which MMP-2 and MMP-9 play significant roles.

## 4. Materials and Methods

### 4.1. Reagents

American Type Culture Collection (ATCC, Rockville, USA) provided the THP-1 cells for the study. Monocytes and macrophages of the cell line THP-1 obtained from the cells serve as a widely used cell model used in investigating immune response. It was found that the results of the studies conducted on the said cells also applied to the human organism. THP-1 showed similarity to the human monocytes/macrophages with respect to morphology and function [13]. It was shown that they provide an appropriate model for the analysis of monocyte/macrophage response with respect to “macrophage differentiation”, and for the purpose of investigating the effect of external factors on macrophages [14]. RPMI culture medium, glutamine and antibiotics (penicillin and streptomycin), phosphate buffered saline (PBS), phorbol 12-myristate 13-acetate (PMA) were purchased from Sigma–Aldrich (Poznań, Poland). Foetal bovine serum was purchased from Gibco (Gibco, Paisley, UK). Metalloproteinase 2 (MMP2) EIA Kit and metalloproteinase 9 (MMP9) EIA Kit were purchased from Cayman, USA whereas Micro BCA Protein Assay kit was obtained from Thermo Scientific (Poznań, Poland).

Micro BCA Protein Assay kit was purchased from Thermo Scientific (Poznań, Poland). Primary monoclonal antibodies against MMP2, MMP9 and β-actin were purchased from Santa Cruz Biotechnology (Heidelberg, Germany). Secondary antibodies (goat anti-mouse IgG HPR) were obtained from Santa Cruz Biotechnology (Heidelberg, Germany). MTA Repair HP was purchased from Septodont (Saint Maur des Fsses, France).

### 4.2. Chemical Properties of MTA Repair HP

As specified in the manufacturer’s data, in comparison with the original MTA, the parameters of MTA Repair HP were modified in terms of its plasticity and ease of placing in the surgical site. This stems from the size of the particles which, following mixing with liquid, provide substantial ease of manipulation. The addition of a new compound CaOW4 allows improved visibility on X-ray images without subsequent iatrogenic discoloration of the root or clinical crown. MTA Repair HP is characterized by low solubility in tissue fluids without any effect on its strength parameters. It provides a superior sealing ability of the sites of iatrogenic perforation which prevents the migration of microorganisms to the treatment area. Such tightness provides reduction of the inflammation. It induces the formation of periradicular cement. Additionally, it allows for dentin bridge formation when it is used for direct pulp capping [26]. Table 2 presents detailed manufacturer data on the use of MTA Repair HP, see also Supplementary data, *2S. Chemical properties of MTA.*

### 4.3. Preparation of MTA Repair HP

First, the equipment needed for mixing and combining the powder with the liquid was sterilized. The content of one package—a single dose of MTA Repair HP powder and 3 drops of liquid were placed onto a glass plate. The mixing lasted 40 s, until the powder and the liquid formed a uniform mass. The consistency of the obtained cement was plasticine-like. The preparation was used immediately for connecting with monocytes of THP-1 cell line.

### 4.4. Cell Culture and Treatment

The experiments were conducted on human monocytes of THP-1 cell line and macrophages obtained from THP-1 cells. THP-1 cell line was obtained originally as a transformed cell line from a patient with acute monocytic leukaemia. This is a commonly employed research model in studies on physiology and pathophysiology of human monocytes and macrophages. In particular, THP-1 cells are used for investigation of the mechanisms of inflammatory reaction [39,40]. On this account, THP-1 monocytes/macrophages were selected as a model for metalloproteinase expression in this study.

The cells were cultured in medium Roswell Park Memorial Institute (RPMI) 1640 (Sigma, St. Louis, MO, USA), supplemented with 100 IU/ml penicillin and 10 µg/mL streptomycin (Life Technologies, Inc., Grand Island, NY), in the presence of 10% thermally inactivated foetal bovine serum (FBS), (Gibco, Paisley, UK). The cells were cultured in a moist atmosphere at the temperature of 37 °C in 5% CO_2_, and the culture medium was refreshed every 48 h. Prior to the experiment, THP-1 cells were placed in culture flasks with the initial density of 2.5 × 10^5^ cells/well (Corning, Cambridge, MA, USA). Differentiation of THP-1 cells into macrophages was obtained by the addition of 10 nM PMA for 24 h [41,42,43].

### 4.5. MTA Repair HP-Induced Inflammatory Reaction in THP-1 Monocytes

In the first experiment, THP-1 cells were grown for 24 and 48 h in RPMI culture medium with 10% FBS, in the presence of MTA Repair HP prepared according the manufacturer’s instructions. Following the incubation period, the cells were harvested by scraping and the pellets were obtained by centrifugation (800 G, 10 min). Next, cold PBS and protease inhibitor cocktail (Merck, Poznań, Poland) was added to cellular sediments and the samples were frozen at −80 °C for further analysis. Next protein concentration was measured using Micro BCA Protein Assay Kit (Thermo Scientific, Rockford, IL, USA). The remaining supernatants were placed in new test tubes and kept at −80 °C for further analysis—measurement of MMT-2 and MMT-9 protein expression using ELISA and Western blot method. Differentiation of monocytes THP-1 to macrophages THP-1 (activation of monocytic cells THP-1) was measured by flow cytometry. The culture model in vitro used in the present study has been previously used for the purpose of clinical verification of the induction of inflammatory reaction of materials based on hydrated calcium silicate with odontotropic properties [33].The control cells were incubated in 10% FBS RPMI medium as control 24 (C 24) for 24 h for 48 h as control 48 (C 48).

### 4.6. Differentiation of THP-1 Cells into Macrophages. Flow Cytometry Measurement

The activity of THP-1 monocytic cells (differentiation in macrophages without PMA and only in the presence of MTA Repair HP, as in the first experiment) was verified by the expression of CD 14 and CD 68 antibodies, as evaluated by flow cytometry using mouse antibody against human CD14 FITC, and mouse antibody against human CD64 Alex Fluor 647 clone Y1: 82A (BD Pharmingen San Diego, CA, USA). The cells were compared to the isotype control, IgG1 κ mouse and IgG2b κ mouse (BD Pharmingen). We typically use 1 × 10^6^ cells and 5 μL CD68/20 uL CD14 in a 100 uL experimental sample. In brief, the procedure consisted of the following: the cells were stained in phosphate buffered saline (PBS, Ca^2+^ and Mg^2+^) supplemented with 2% bovine calf serum (BCS, Hyclone, Logan, UT, USA). After the last wash, the cells were re-suspended in PBS and analyzed with FACS using a Navios flow cytometer (Beckman Coulter, Brea, CA, USA).

### 4.7. MTA Repair HP-Induced Inflammatory Reaction in Macrophages

In the course of the second experiment, THP-1 macrophages were cultured for 24 and 48 h under the same conditions as in the first experiment with the presence of MTA Repair HP. Following incubation, the cells were harvested as previously, and the protein concentration was measured using the same method as in the first experiment—Micro BCA Protein Assay Kit (Thermo Scientific, Rockford, IL, USA); MMT-2 and MMT-9 protein expression was measured with ELISA and Western blot method, just as in the first experiment, and later visualised in confocal microscopy.

### 4.8. Determination of MMP-2 Expression by ELISA Method

The expression of MMP-2 was analyzed using commercially available quantitative test ELISA (Thermo Fisher Scientific, MMP-2 Human ELISA Kit, KHC3081) following the manufacturer’s instructions.

### 4.9. The Determination of MMP-9 Expression by ELISA Method

The expression of MMP-9 was analyzed using commercially available quantitative test ELISA (Thermo Fisher Scientific, MMP-9 Human ELISA Kit, BMS2016-2) following the manufacturer’s instructions.

### 4.10. Western Blot Analysis of MMP-2 and MMP-9 Expression

The cells incubated with MTA Repair HP were rinsed with PBS. After scraping, the cells were lysed with protease inhibitor lysis buffer, ethylene-diaminetetra-acetic acid 5 mM, sodium dichloroisocyanurate 1%, TRITON-X 1% and equal amounts of protein (20 ug) were separated in gel electrophoresis and then transferred to a nitrocellulose membrane (Thermo Scientific, Pierce Biotechnology, Rockford, IL, USA) at 157 mA for 2 h at room temperature. After blocking the membrane of 5% non-fat milk in Tris-buffered saline (Sigma-Aldrich, Poland) containing 0.1% Tween 20 (Sigma-Aldrich, Poland) for the whole night at 4 °C, there was an incubation with primary monoclonal antibodies direct against MMP2 (cat. No.: sc-13594) or MMP9 (cat. No.: sc-393859) Santa Cruz Biotechnology, USA) in a dilution of 1:200 with a monoclonal anti-ß-actin (1:5000; Santa Cruz Biotechnology, USA) and next with secondary antibodies (goat anti-mouse IgG HRP; cat. No. sc-2031; Santa Cruz Biotechnology, USA) in a dilution of 1:5000. The signals were visualized by chemiluminescence (Thermo Scientific, Pierce Biotechnology, Rockford, IL, USA). ImageJ 1.41o (NIH, USA) was used to densitometric analysis of bands.

### 4.11. Confocal Microscopy Imaging of MMP-2 and MMP-9 Expression

The expression of MMP-2 and MMP-9 proteins was examined with confocal microscopy. THP-1 macrophages were grown on cover glasses under standard in vitro culture conditions. Then, the cells were washed with PBS and fixed with 4% buffered formalin for 15 min at room temperature (RT). After fixation and washing with PBS, the cells were permeabilized with 0.5% solution of Triton X-100 in PBS. After washing with fresh portion of PBS, the cells were incubated with primary antibodies: mouse anti-MMP2 and mouse anti-MMP9 (Santa Cruz Biotechnology, Heidelberg, Germany) in 1:50 dilution, at 4 °C overnight and then washed and incubated with the secondary antibody: anti mouse IgG FITC conjugated, dilution 1:60 (Sigma-Aldrich, Poznań, Poland) in antibody diluent (Dako), 30 min in RT and, after washing with PBS, further with Hoechst 33258, 30 min., RT. The cells were examined under a confocal microscope (FV1000 confocal with IX81 inverted microscope, Olympus, Germany). Three channel acquisition and sequential scanning were used for best resolution of signal from Hoechst 33,258 and FITC fluorescence. Additionally, fluorescent images were merged with transition light images.

In order to quantitative evaluation of enzyme expression cells were incubated with antibodies at the same concentration and condition of microscopic studies and washed three times with PBS at room temperature. Fluorescence intensity was measured in microplates dedicated for fluorescent studies (Eppendorf) by Asys UVM 340 microplate reader (AsysHitechGmbh, Austria) at the wavelength specified above. Fluorescence was normalized to protein levels measured by Micro BCA™ Protein Assay Kit (Thermo Scientific, Pierce Biotechnology, Rockford, IL, USA).

### 4.12. Protein Concentration

Protein concentration was measured with a MicroBCA™ Protein Assay Kit (Thermo Scientific, Pierce Biotechnology, USA) using a spectrophotometer (UVM340, ASYS).

### 4.13. Statistical Analysis

The statistical analysis of the results obtained in the course of the present study was conducted with the use of Statistica 10 software (Statsoft, Poznań, Poland). The results are expressed as arithmetical means ± standard deviation (SD).The distribution of variables was evaluated with a Shapiro–Wilk W-test.For the purpose of further analyses, nonparametric tests were used since the distribution, in most cases, deviated from the normal distribution. The results were subjected to Mann–Whitney U test. The significance level was set at <0.05.

## 5. Conclusions

The results of our study confirm that mineral trioxide aggregate (MTA Repair HP) does not alter MMP-2 and MMP-9 protein expression in the cultured monocytes/macrophages.

## Figures and Tables

**Figure 1 ijms-22-00295-f001:**
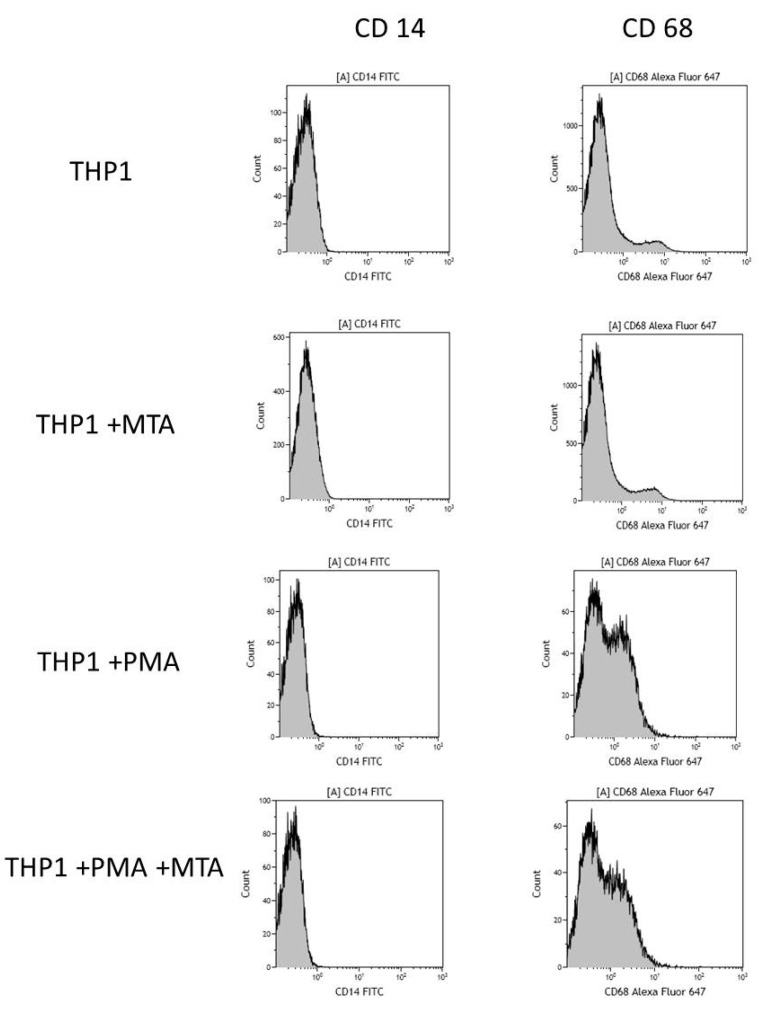
The effect of mineral trioxide aggregate Repair HP (MTA) on the activation of THP-1 monocytic cells. THP-1 monocytic cells were incubated in the presence of MTA Repair HP without PMA (upper) and treated with PMA (lower). Expression was determined by flow cytometry. CD68 expression (marker for macrophage differentiation) increased after 48 h treatment with PMA (200 nM) as compared to non-treated cells. Expression of CD14 (marker for monocyte differentiation) did not change. Experiments were conducted as six separate assays.

**Figure 2 ijms-22-00295-f002:**
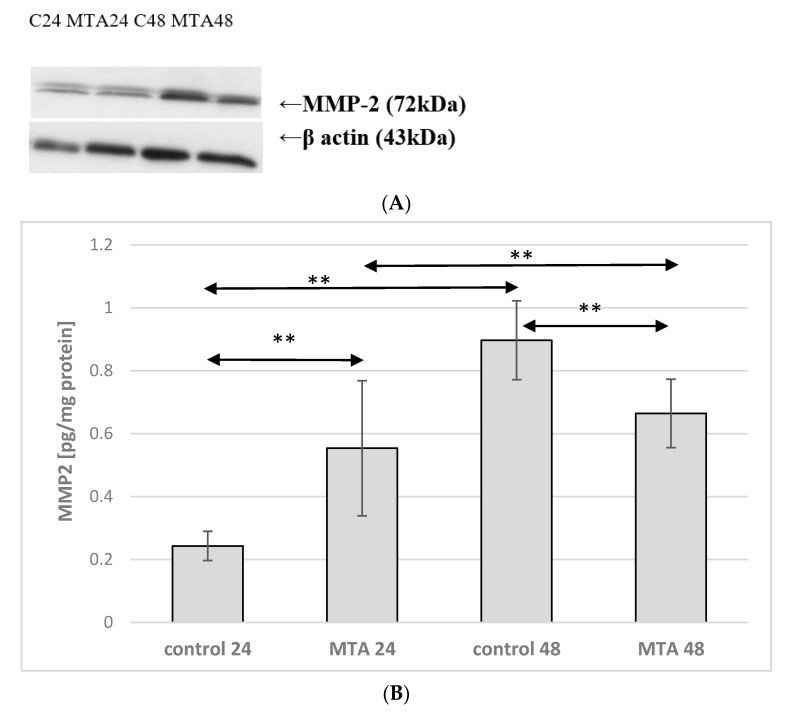
MMP-2 protein expression in monocytes. (**A**) Representative enzyme protein expression (70 kDa) normalized to β-actin determined by Western blotting method was shown (**B**) ELISA method analysis of enzyme protein in monocytes was shown. Cells were cultured with mineral trioxide aggregate (MTA Repair HP) for 24 h (MTA24) and 48 h (MTA48) in RPMI medium with 10% FBS. Following incubation, the cells were collected by scraping and protein concentration was measured. The control cells were incubated in 10% FBS RPMI medium for 24 h—control 24 (C 24) or for 48 h—control 48 (C 48). The experiments were conducted as three separate assays. ** statistically significant differences vs. control group, *p* ≤ 0.001; (U–Mann Whitney test).

**Figure 3 ijms-22-00295-f003:**
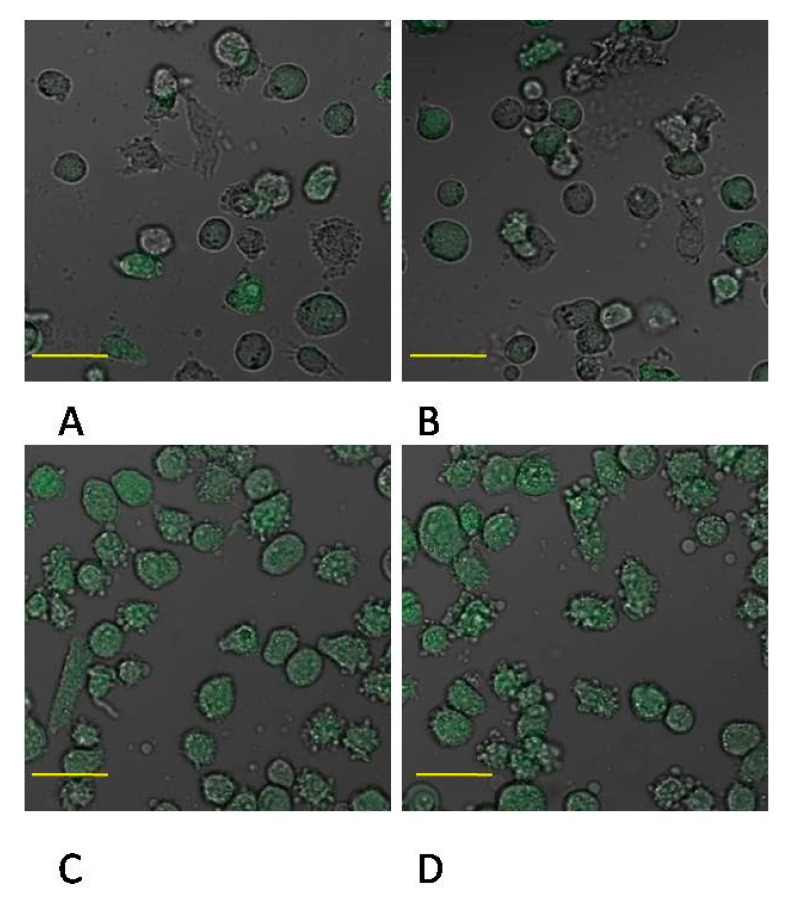
Imaging of MMP-2 enzyme expression by fluorescence microscopy in THP-1 monocytes cultured with mineral trioxide aggregate (MTA Repair HP). Cells were cultured in RPMI medium with 10% FBS—control conditions for 24 h (**A**) and for control conditions for 48 h (**B**); THP1-monocytes cultured in RPMI medium supplemented with 10% FBS with MTA Repair HP for 24 h (**C**) and for 48 h (**D**). Immunofluorescence analysis was performed with the use of specific primary antibody, mouse anti-MMP-2 (overnight incubation at 4 °C) and secondary antibodies conjugated with fluorochrome—anti-mouse IgG-FITC (Fluorescein Isothiocyanate), (incubation for 45 min at room temperature), enzyme expression is visible as green fluorescence and indicates MMP-2 expression. Image analysis was performed using a confocal microscope with filters 38 HE GFP for green fluorescence. The control cells were incubated in 10% FBS RPMI medium as control 24 (C 24) for 24 h for 48 h as control 48 (C 48). The experiments were conducted as six separate assays (each assay in three replicates). There were no differences in enzymes protein expression (*p* ≤ 0.5, U Mann–Whitney test) examined by measuring the intensity of fluorescence through a plate reader (data not shown). Scale bar 40 μm.

**Figure 4 ijms-22-00295-f004:**
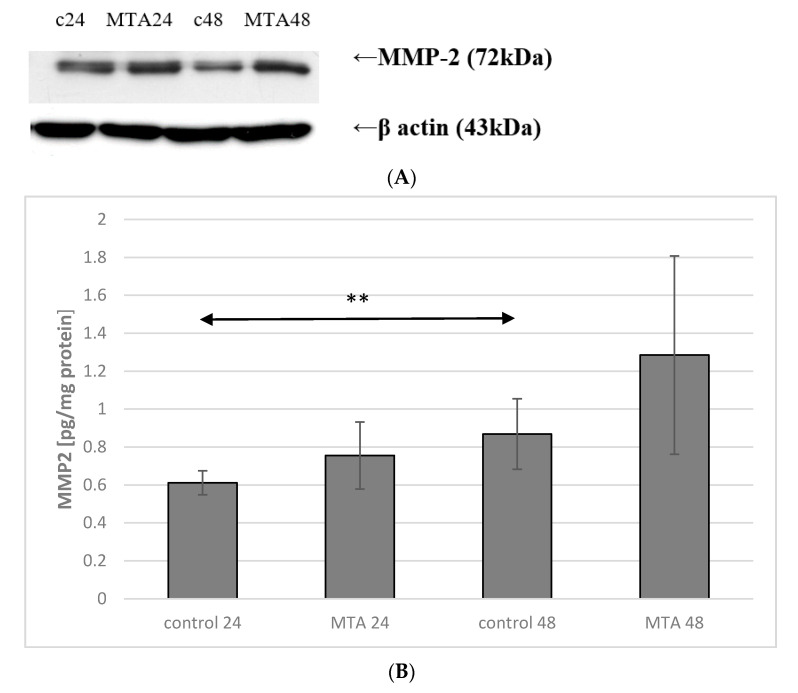
MMP-2 protein expression in macrophages. (**A**) Representative enzyme protein expression (70 kDa) normalized to β-actin determined by Western blotting method was shown (**B**) ELISA method analysis of enzyme protein (**B**) in macrophages was shown. Cells were cultured with mineral trioxide aggregate (MTA Repair HP) for 24 h (MTA24) and 48 h (MTA48) in RPMI medium with 10% FBS. Following incubation, the cells were scraped and protein concentration was measured. The control cells were incubated in 10% FBS RPMI medium for 24 h—control 24 (C 24) or for 48 h—control 48 (C 48). The experiments were conducted as three separate assays. ** statistically significant differences vs. control group, *p* ≤ 0.001; (U–Mann Whitney test).

**Figure 5 ijms-22-00295-f005:**
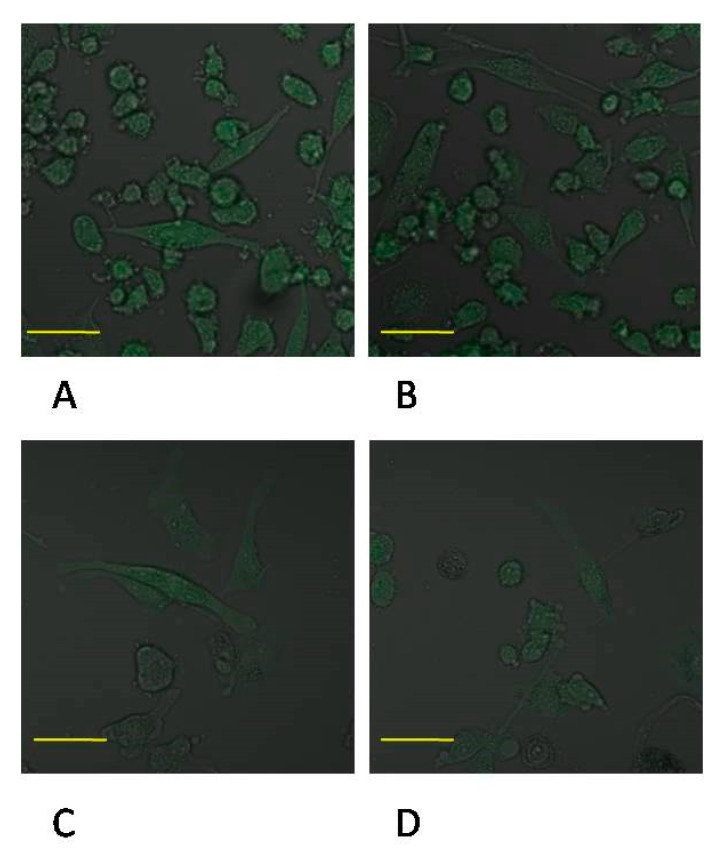
Imaging of MMP-2 enzyme expression by fluorescence microscopy in macrophages cultured with mineral trioxide aggregate (MTA Repair HP). Cells were cultured in RPMI medium with 10% FBS—control conditions for 24 h (**A**) and for control conditions for 48 h (**B**); THP1-monocytes cultured in RPMI medium with 10% FBS with MTA Repair HP for 24 h (**C**) and for 48 h (**D**). Immunofluorescence was performed with the use of specific primary antibody, mouse anti-MMP-2 (overnight incubation at 4 °C) and secondary antibodies conjugated with fluorochrome—anti-mouse IgG-FITC (incubation for 45 min at room temperature), MMP-2 expression is indicated by the visible green fluorescence. Image analysis was performed using a confocal microscope with 38 HE GFP filters for green fluorescence. The control cells were incubated in 10% FBS RPMI medium as control 24 (C 24) for 24 h for 48 h as control 48 (C 48). The experiments were conducted as six separate assays (each assay in three replicates). There were no differences in enzymes protein expression (*p* ≤ 0.5, U Mann–Whitney test) examined by measuring the intensity of fluorescence through a plate reader (data not shown). Scale bar 40 μm.

**Figure 6 ijms-22-00295-f006:**
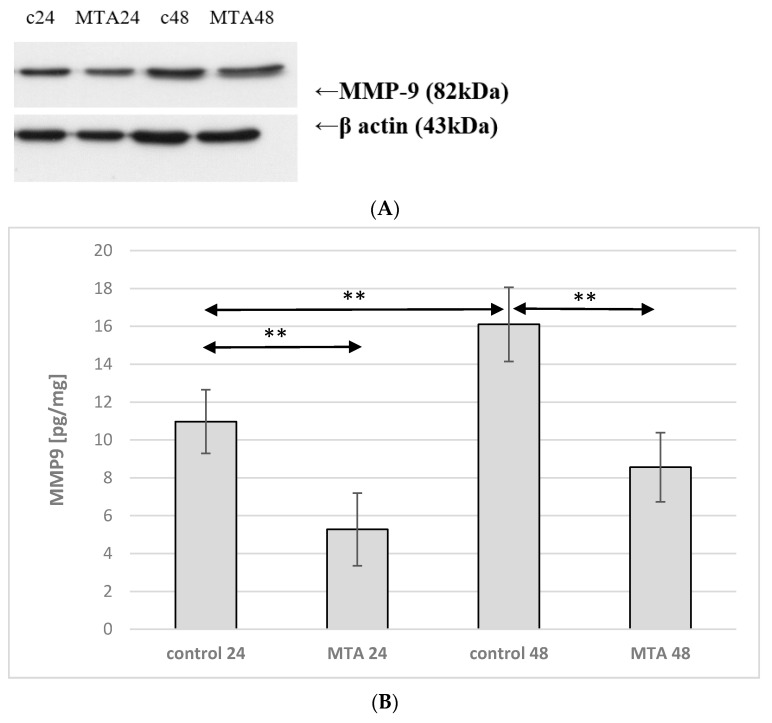
MMP-2 protein expression in monocytes. (**A**) Representative enzyme protein expression (70 kDa) normalized to β-actin determined by Western blotting method was shown (**B**) ELISA method analysis of enzyme protein (**B**) in monocytes was shown. Cells were cultured with mineral trioxide aggregate MTA Repair HP for 24 h (MTA24) and 48 h (MTA48) in RPMI medium with 10% FBS. Following the incubation period, the cells were scraped and protein concentration was measured. The control cells were incubated in RPMI medium with the addition of 10% FBS for 24 h constituting the control 24 (C 24) or for 48 h—control 48 (C 48). The experiments were conducted as three separate assays. ** statistically significant differences vs. control group, *p* ≤ 0.001; (U–Mann Whitney test).

**Figure 7 ijms-22-00295-f007:**
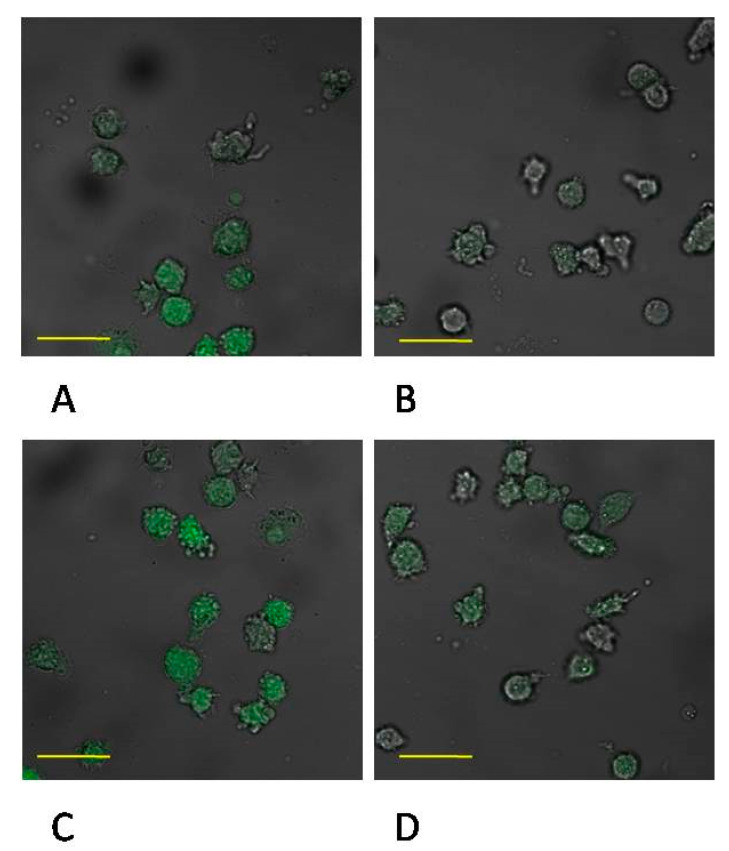
Imaging of MMP-9 enzyme expression by fluorescence microscopy in THP-1 monocytes cultured with mineral trioxide aggregate Repair HP (MTA Repair HP). Cells were cultured in RPMI medium with 10% FBS—control conditions for 24 h (**A**) and for 48 h (**B**); THP1-monocytes cultured in 10% FBS RPMI medium with MTA Repair HP for 24 h (**C**) and 48 h (**D**). Immunofluorescence was performed with specific primary antibody, mouse anti-MMP-2 (overnight incubation at 4 °C) and secondary antibodies conjugated with fluorochrome—anti-mouse IgG-FITC (incubation for 45 min at room temperature), enzyme expression manifested by green fluorescence indicates expression of MMP-2. Image analysis was performed using a confocal microscope with 38 HE GFP filters for green fluorescence. The control cells were incubated in 10% FBS RPMI medium as control 24 (C 24) for 24 h for 48 h as control 48 (C 48). The experiments were conducted as six separate assays (each assay in three replicates). There were no differences in enzymes protein expression (*p* ≤ 0.5, U Mann–Whitney test) examined by measuring the intensity of fluorescence through a plate reader (data not shown). Scale bar 40 μm.

**Figure 8 ijms-22-00295-f008:**
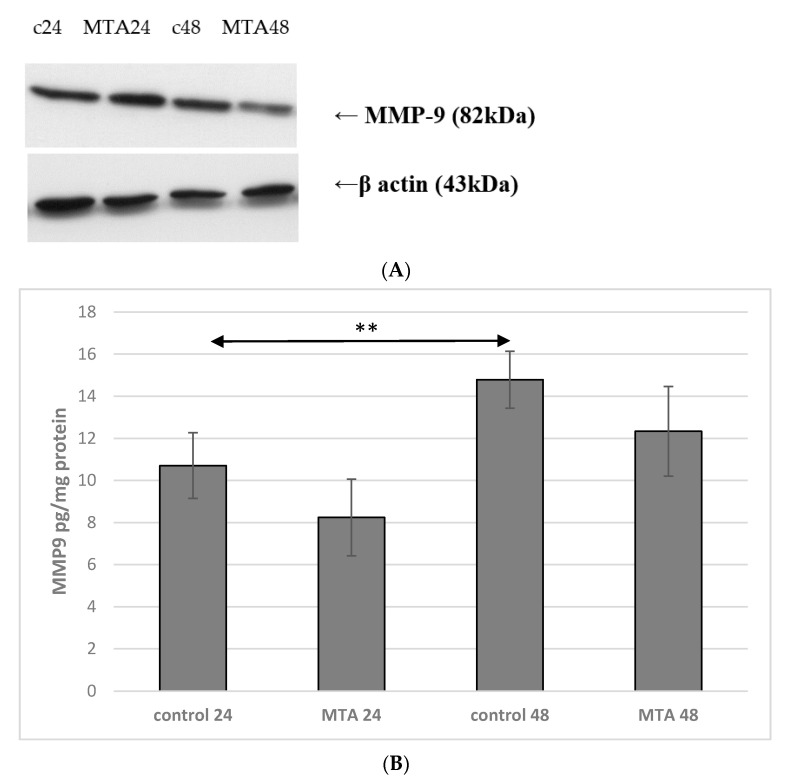
MMP-9 protein expression in macrophages. (**A**) Representative enzyme protein expression (70 kDa) normalized to β-actin determined by Western blotting method was shown (**B**) ELISA method analysis of enzyme protein (**B**) in macrophages was shown. The cells were cultured with mineral trioxide aggregate (MTA Repair HP) for 24 h (MTA24) and 48 h (MTA48) in RPMI medium with the addition of 10% FBS. Following incubation time, the cells were collected by scraping and protein concentration was measured. The control cells were incubated in 10% FBS RPMI medium as control 24 (C 24) for 24 h for 48 h as control 48 (C 48). The experiments were conducted as three separate assays. ** statistically significant differences vs. control group, *p* ≤ 0.001; (U–Mann Whitney test).

**Figure 9 ijms-22-00295-f009:**
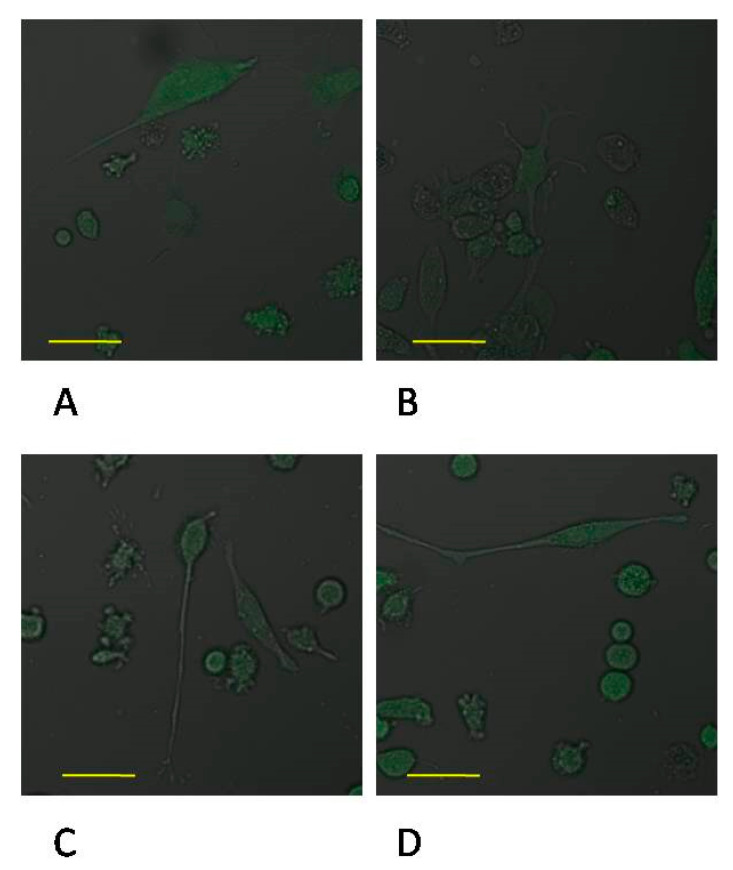
Imaging of MMP-9 enzyme expression with fluorescence microscopy in macrophages cultured with mineral trioxide aggregate (MTA Repair HP). The cells were cultured in RPMI medium with 10% FBS as control conditions for 24 h (**A**) and as control conditions for 48 h (**B**); THP1-monocytes cultured in RPMI medium with 10% FBS with MTA Repair HP for 24 h (**C**) and for 48 h (**D**). Immunofluorescence was performed with specific primary antibody, mouse anti-MMP-2 (overnight incubation at 4 °C) and secondary antibodies conjugated with fluorochrome—anti-mouse IgG-FITC (incubation for 45 min at room temperature), MMP-2 expression is manifested by green fluorescence. Image analysis was performed by means of confocal microscope with 38 HE GFP filters for green fluorescence. The control cells were incubated in 10% FBS RPMI medium as control 24 (C 24) for 24 h for 48 h as control 48 (C 48). The experiments were conducted as six separate assays (each assay in three replicates). There were no differences in enzymes protein expression (*p* ≤ 0.5, U Mann–Whitney test) examined by measuring the intensity of fluorescence through a plate reader (data not shown). Scale bar 40 μm.

**Table 1 ijms-22-00295-t001:** The effect of MTA Repair HP on THP-1 monocytes activation. THP-1 monocytic cells were incubated in the presence of MTA Repair HP, without PMA or treated with PMA. The expression of CD 68 receptor (a marker for macrophage differentiation) or CD 14 receptor (a marker for monocyte differentiation) was determined by flow cytometry. The experiments were conducted as six separate assays (each assay in three replicates). ns—expression of CD14 in cells showed no significant changes, *p* = 0.65 (U Mann–Whitney test). ** statistically significant differences vs. control group, *p* ≤ 0.001; (U-Mann Whitney test). ## statistically significant differences vs. MTA Repair HP group, *p* ≤ 0.001; (U-Mann Whitney test).

	Flow Cytometry	
	CD14 Expression	CD68 Expression
	% Positive Cells	
THP1	0.8	2.9
THP1 + MTA Repair HP	0.7	2.7
THP1 + PMA	0.1 ^ns^	45.2 **
THP1 + PMA + MTA Repair HP	0.1 ^ns^	43.5 ^##^

**Table 2 ijms-22-00295-t002:** Detailed manufacturer data on the MTA Repair HP used for testing.

Preparation	Producer	Composition
MTA RepairHP	Angelus, Londrina, Brasil	Powder: Tricalcium Silicate 3CaO·SiO_2,_ Dicalcium Silicate 2CaO·SiO_2,_ Tricalcium Aluminate 3CaO·Al_2_O_3,_ Calcium Oxide CaO, Calcium Tungstate CAWO_4_ Liquid: Water and Plasticizer

## Data Availability

All raw data used in this manuscript has been deposited in the depository of the Department of Biochemistry, Pomeranian Medical University, available from the corresponding author.

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
