# Peer review of "The Influence of New Silicate Cement Mineral Trioxide Aggregate (MTA Repair HP) on Metalloproteinase MMP-2 and MMP-9 Expression in Cultured THP-1 Macrophages"

_ijms, 2020, doi:10.3390/ijms22010295_

Round 1

Reviewer 1 Report

I think Authors have made a very good job.

Author Response

We greatly appreciate for review, thank you.

Reviewer 2 Report

This study explores a very relevant topic in clinical practice, that of the use of calcium silicate-based cements. In fact, there is an    increase  in  the  clinical  applications of  bioceramics in restorative  dentistry  as well as in endodontics. For example, this family of materials are important in vital pulp therapy as they have been shown to induce pup repair. But it is also important that these materials used for pulp repair do not cause additional inflammation (in addition to residual pulp inflammation) and do  not  interfere  with  the  regeneration  process.

The rationale of the current study is interesting. Indeed, seeing the effect of a material such as MTA repair HP on monocytes / macrophages and on matrix metalloproteases could make it possible to know their inflammatory or anti-inflammatory effect. Is it possible to have this data?

Some points should be noted. Indeed, MMPs are detected by WB which gives information about their presence or not. But MMPs are enzymes and it is very important to have information on their enzymatic activity. It would therefore be interesting to carry out zymography to know the enzymatic activity.

It would have been really relevant to supplement these data with an in vivo study. In vitro information is of course interesting, but when it comes to clinical application, relying on preclinical data on animals seems of great importance. Would there be a project in this direction?

Author Response

Review 2

 Comments and Suggestions for Authors

This study explores a very relevant topic in clinical practice, that of the use of calcium silicate-based cements. In fact, there is an  increase  in  the  clinical  applications of  bioceramics in restorative  dentistry  as well as in endodontics. For example, this family of materials are important in vital pulp therapy as they have been shown to induce pup repair. But it is also important that these materials used for pulp repair do not cause additional inflammation (in addition to residual pulp inflammation) and do  not  interfere  with  the  regeneration  process. The rationale of the current study is interesting. Indeed, seeing the effect of a material such as MTA repair HP on monocytes / macrophages and on matrix metalloproteases could make it possible to know their inflammatory or anti-inflammatory effect. Is it possible to have this data?

Thank you for very much for this remark, however, at the present time we can not turn on additional analyzes on the anti-inflammatory agents due to no possiblity to purchase new reagents and antibodies that were not provided in the project. However, it is an excellent direction for our research and we will try to extend it as suggested by the reviewer.

Some points should be noted. Indeed, MMPs are detected by WB which gives information about their presence or not. But MMPs are enzymes and it is very important to have information on their enzymatic activity. It would therefore be interesting to carry out zymography to know the enzymatic activity.

It would have been really relevant to supplement these data with an in vivo study.

We are very grateful for this remark. According to reviewer's remark, we extended the research to include the activity of the tested enzymes using the zymography method. According to reviewer's remark, we extended the research to include the activity of the tested enzymes using the zymography method. We included the data as a supplement to the manuscript.

In vitro information is of course interesting, but when it comes to clinical application, relying on preclinical data on animals seems of great importance. Would there be a project in this direction?

We are very grateful for this remark. According to reviewer's remark we added the use of MTA in patients. We included the data as a supplement to the manuscript.

This manuscript is a resubmission of an earlier submission. The following is a list of the peer review reports and author responses from that submission.

Round 1

Reviewer 1 Report

Dear Authors,

this excellent paper is very interesting for both, the scientists and clinicians. I found it valuable for the content of the International Journal of Molecular Sciences.

There are only few editiorial corrections needed to perform:

  1. line nr 51 "gold standard" -> "golden standard"
  2. Fig. 1 D - the arrow is necessary to show where is a perforation because not every reader is a dentist to see it.
  3. line 84 "endodontic disease" - is not commonly used. It's better to exchange it.
  4. Chapter Results. It's worth to mention at the begining that the short term MTA will be used from now instead of MTA Repair HP.
  5. Discussion. Because of there are so many versions of MTA, maybe where is possible authors could add the type of discussed MTA.
  6. line 368 - nr 33 from References list and cited in discussion concerns another silicate cement Biodentine not MTA as described.

Author Response

Comments and Suggestions for Authors

Dear Authors,

this excellent paper is very interesting for both, the scientists and clinicians. I found it valuable for the content of the International Journal of Molecular Sciences.

There are only few editiorial corrections needed to perform:

line nr 51 "gold standard"->"golden standard"

According to the Reviewer's remark we corrected the sentence.

Fig. 1 D - the arrow is necessary to show where is a perforation because not every reader is a dentist to see it.

According to the Reviewer's remark we corrected Fig.1 and added arrows to show a location of the perforation (Fig.1 A, B, C, D, E) and of the perforation repaired with MTA (Fig.1 F, G, H, I).

line 84 "endodontic disease"- is not commonly used. It's better to exchange it “periodontitis”

According to the Reviewer's remark we corrected the sentence.

Chapter Results. It's worth to mention at the beginning that the short term MTA will be used from now instead of MTA Repair HP.

According to the Reviewer's remark we added this remark in this chapter

Discussion. Because of there are so many versions of MTA, maybe where is possible authors could add the type of discussed MTA.

The types of MTAs were not given in every publication. Whenever possible, the type of MTA covered by the description was given in the discussion.

line 368 - nr 33 from References list and cited in discussion concerns another silicate cement Biodentine not MTA as described.

According to the Reviewer's remark we corrected this sentence.

Reviewer 2 Report

I find the paper so new and interesting. My congratulations to the authors!

Author Response

Review 2

Comments and Suggestions for Authors

I find the paper so new and interesting. My congratulations to the authors!

We are very grateful for this Review, thank you.

Reviewer 3 Report

In the manuscript entitled “The influence of New Silicate Cement Mineral Trioxide Aggregate (MTA Repair HP) on Metalloproteinase MMP-2 and MMP-9 expression in cultured THP-1 Macrophages”, Katarzyna Barczak and colleagues investigated the relationship between the new silicate cement Mineral Trioxide Aggregate (MTA Repair HP) and the inflammation process.

The aim of the study is to demonstrate that the endodontic cement MTA Repair HP does not increase inflammatory response in vitro.

To reach this aim the authors performed in vitro analyses using THP-1 cells.

The topic of the work is interesting; however, the study is still too preliminary to be published in the present form. It is hard to justify the conclusion with the methodology chosen and the results obtained therefrom.

From my point of view, the manuscript requires a general revision and there are several important aspects needing the authors’ attention.

1) The text is often redundant. Too much space is given to information of little relevance to the present study. The authors should remove from the manuscript all the sentences that represent a mere description of MTA material and of its use in clinical practice. The article should contain only the information necessary for the reader to understand the purpose of the work.

2) The methods section needs to be completely revised. Only the information necessary to understand the employed procedures and to ensure the reproducibility of the experiments should be included. 

At the same time, a lot of important information is missing. For example, the authors did not indicate the concentrations of antibodies for cytometry tests.

The authors did not describe how the cells were evaluated through confocal microscopy. What parameters were used? How were they quantified?

The authors did not indicate the amount of protein used for western blot analysis.

The controls used in the different procedures are not clearly stated.

3) Going through the methods it is not clear which tests were conducted on monocytes and which on stimulated monocytes (treated with PMA). Please specify this point.

4) It is not clear why cells were frozen in PBS. The authors should explain how they avoided cell damage during the freezing/thawing phase.

5) The different sections described in the methods intersect each other. The authors should reorganize the text more clearly. For example, it is not clear why in section 4.4 the authors wrote "Differentiation of monocytes THP-1 to macrophages THP-1 (activation of monocytic cells THP-1) was measured by flow cytometry." if then section 4.4.1 is related to "Differentiation of THP-1 cells into macrophages. Flow cytometry measurement".

6) The procedure used by the authors to prepare MTA is absolutely unclear. Since the results of the study depend on this factor, it is mandatory to understand the procedure used.

7) Why did the authors not include a comparison between MTA Repair HP and the other MTA formulations? It should be done.

8) The authors should include flow-cytometry plots in the results section to support text and data.

9) In the results section the authors often mention "control condition", but in the methods section it is not clearly described what it corresponds to.

10) In the results section the authors report that they “have/have not” detected differences using microscopy. First of all, it is not clear how they have quantified these differences.

Furthermore, they should state at what magnification the images were obtained.

In addition, looking at the figures, it can be seen that cells have different morphologies. The authors did not comment on this. Can morphological changes be due to the presence of MTA? Please clarify this point and discuss these results.

11) The authors should discuss the increases detected in their control cells.

12) It would be interesting if the authors also included co-culture tests with monocytes/macrophages and an osteoblastic-like cell line to reproduce an in vitro model of a real-life condition.

13) Whole of discussion is poorly written and insufficient. The authors should improve the comments related to their results and should expanded the text with comparisons in details to previous literature.

14) It is not clear why the authors have included Figure 1. Figure 1 does not include data related to the experiments performed in this study. The authors should better motivate the presence of this Figure, otherwise they should remove it.

Minor

a) Figures are not cited in the text.

b) The authors improperly use the term “immunohistochemistry” in the text.

c) The English language needs thorough revision by an English native speaker.

Author Response

Review 3

Comments and Suggestions for Authors

In the manuscript entitled “The influence of New Silicate Cement Mineral Trioxide Aggregate (MTA Repair HP) on Metalloproteinase MMP-2 and MMP-9 expression in cultured THP-1 Macrophages”, Katarzyna Barczak and colleagues investigated the relationship between the new silicate cement Mineral Trioxide Aggregate (MTA Repair HP) and the inflammation process.

The aim of the study is to demonstrate that the endodontic cement MTA Repair HP does not increase inflammatory response in vitro.

To reach this aim the authors performed in vitroanalyses using THP-1 cells.

The topic of the work is interesting; however, the study is still too preliminary to be published in the present form. It is hard to justify the conclusion with the methodology chosen and the results obtained therefrom.

From my point of view, the manuscript requires a general revision and there are several important aspects needing the authors’ attention.

1) The text is often redundant. Too much space is given to information of little relevance to the present study. The authors should remove from the manuscript all the sentences that represent a mere description of MTA material and of its use in clinical practice. The article should contain only the information necessary for the reader to understand the purpose of the work.

We are grateful for this comment, however we think that information given in the manuscript is indispensable for proper understanding of the issue both in terms of clinical as well as biochemical aspects. MTA is a new material and, in our opinion, providing its detailed characteristics is crucial. Also, we would like to emphasize that other Reviewers did not raise doubts regarding the said issue. Therefore, we would like to keep the manuscript in the present form.

2) The methods section needs to be completely revised. Only the information necessary to understand the employed procedures and to ensure the reproducibility of the experiments should be included. 

At the same time, a lot of important information is missing. For example, the authors did not indicate the concentrations of antibodies for cytometry tests.

Thank you for this remark, we also believe that the manuscript should provide only the necessary information needed for understanding the applied procedures and ensure repeatability of the experiments. Therefore, according to the Reviewer's comments, the Material and methods section was edited.

We typically use 1x106 cells and 5uL CD68/20uL CD14 in a 100uL experimental sample.

The authors did not describe how the cells were evaluated through confocal microscopy. What parameters were used? How were they quantified?

According to the Reviewer's remark we have concluded the method section with an explanation of the confocal microscopy method and quantification

4.9 Confocal microscopy imaging of MMP-2 and MMP-9 expression

The expression of MMP-2 and MMP-9 proteins was examined with confocal microscopy. THP-1 macrophages were grown on cover glasses under standard in vitro culture conditions. Then, the cells were washed with PBS and fixed with 4% buffered formalin for 15 minutes at room temperature (RT). After fixation and washing with PBS, the cells were permeabilized with 0.5% solution of Triton X-100 in PBS. After washing with fresh portion of PBS, the cells were incubated with primary antibodies: mouse anti-MMP2 and mouse anti-MMP9 (Santa Cruz Biotechnology, Heidelberg, Germany) in 1:50 dilution, at 4oC overnight and then washed and incubated with the secondary antibody: anti mouse IgG FITC conjugated, dilution 1:60 (Sigma-Aldrich, Poznań, Poland) in antibody diluent (Dako), 30 minutes in RT and, after washing with PBS, further with Hoechst 33258, 30 min., RT. The cells were examined under a confocal microscope (FV1000 confocal with IX81 inverted microscope, Olympus, Germany). Three channel acquisition and sequential scanning were used for best resolution of signal from Hoechst 33258 and FITC fluorescence. Additionally, fluorescent images were merged with transition light images.

In order to quantitatively evalevaluate enzyme expression, cells were incubated with antibodies at the same concentration, same conditions of microscopic studies and the cells were washed three times with PBS at room temperature. Fluorescence intensity was measured in microplates dedicated for fluorescent studies (Eppendorf) with Asys UVM 340 microplate reader (Asys Hitech Gmbh, Austria) at the wavelength specified above. Fluorescence was normalized to protein levels measured by Micro BCA™ Protein Assay Kit (Thermo Scientific, Pierce Biotechnology, USA.

Moreover, we would like to point out that we have conducted an extremely detailed assessment of quantitative expression of the analysed metaloproteinases with ELISA (giving the amount in pg/mg proteing in figures and charts - Fig. 2B, 4B, 6B, 8B).

The authors did not indicate the amount of protein used for western blot analysis.

According to the Reviewer's remark we have concluded the method section with the amount of protein used for western blot analysis.

4.8. Western Blot analysis of MMP-2 and MMP-9 expression

The cells incubated with MTA were rinsed with PBS. After scraping, the cells were lysed with protease inhibitor lysis buffer, ethylene-diaminetetra-acetic acid 5 mM, sodium dichloroisocyanurate 1 %, TRITON-X 1 % and equal amounts of protein (20 ug) were separated in gel electrophoresis and then transferred to a nitrocellulose membrane (Thermo Scientific, Pierce Biotechnology, USA) at 157 mA for 2 hours at room temperature. After blocking the membrane of 5 % non-fat milk in Tris-buffered saline (Sigma-Aldrich, Poland) containing 0.1 % Tween 20 (Sigma-Aldrich, Poland) for the whole night at 4°C, there was an incubation with primary monoclonal antibodies direct against MMP2 (cat. No.: sc-13594) or MMP9 (cat. No.: sc-393859) Santa Cruz Biotechnology, USA) in a dilution of 1:200 with a monoclonal anti-ß-actin (1:5000; Santa Cruz Biotechnology, USA) and next with secondary antibodies (goat anti-mouse IgG HRP; cat. No. sc-2031; Santa Cruz Biotechnology, USA) in a dilution of 1:5000. The signals were visualized by chemiluminescence (Thermo Scientific, Pierce Biotechnology, USA). ImageJ 1.41o (NIH, USA) was used to densitometric analysis of bands.

The controls used in the different procedures are not clearly stated.

According to the Reviewer's remark we have concluded the results section with the control condition

“The control cells were incubated in 10% FBS RPMI medium as control 24 (C 24) for 24h for 48h as control 48 (C 48)”.

3) Going through the methods it is not clear which tests were conducted on monocytes and which on stimulated monocytes (treated with PMA). Please specify this point.

All tests were performed on monocytes and then on macrophages as described in detail in the Materials and methods section

4.4. MTA-induced inflammatory reaction in THP-1 monocytes

In the first experiment, THP-1 cells….

In the course of the second experiment, THP-1 macrophages

4.5 MTA induced inflammatory reaction in macrophages

4) It is not clear why cells were frozen in PBS. The authors should explain how they avoided cell damage during the freezing/thawing phase.

This is a standard preparation of samples for further biochemical determination (protein measurement, ELISA and Western Blotting method). The cells were not used for culture further.

Cold PBS was added to cellular sediments and the samples were frozen at -80°C for further analysis.

5) The different sections described in the methods intersect each other. The authors should reorganize the text more clearly. For example, it is not clear why in section 4.4 the authors wrote "Differentiation of monocytes THP-1 to macrophages THP-1 (activation of monocytic cells THP-1) was measured by flow cytometry."if then section 4.4.1 is related to "Differentiation of THP-1 cells into macrophages. Flow cytometry measurement".

All analyses were carried out on monocytes and macrophages. To make sure, whether THP-1 monocytes transformed into macrophages (due to PMA or analysed MTA), we studies the expression of CD68 receptor (a marker of macrophages). Therefore, on our opinion, the order and structure of the sub-chapters (4.4 followed by 4.4.1) is reasonable.

4.4 MTA-induced inflammatory reaction in THP-1 monocytes

4.4.1 Differentiation of THP-1 cells into macrophages. Flow cytometry measurement

6) The procedure used by the authors to prepare MTA is absolutely unclear. Since the results of the study depend on this factor, it is mandatory to understand the procedure used.

According to the Reviewer's recommendations, we added the description of the preparation process and mixing of MTA HP Repair (line 440-445).

4.2.1. Prepatiaton of MTA

First, the equipment needed for mixing and combining the powder with the liquid was sterilised. The content of one package - a single dose of MTA Repair HP powder and 3 drops of liquid were placed onto a glass plate. The mixing lasted 40 seconds, until the powder and the liquid formed a uniform mass. The consistency of the obtained cement was plasticine-like. The preparation was used immediately for connecting with monocytes of THP-1 cell line.

Why did the authors not include a comparison between MTA Repair HP and the other MTA formulations? It should be done.

We are very grateful for the comment, however, our study focused entirely on the said type of MTA HP Repair - a new and expensive material. The suggestion to compare the said material with other formulations will be addressed by us in our future research.

8) The authors should include flow-cytometry plots in the results section to support text and data.

According to the Reviewer's remark we added flow cytometry plots.

9) In the results section the authors often mention "control condition", but in the methods section it is not clearly described what it corresponds to.

According to the Reviewer's remark we have concluded the methods section with the control condition

10) In the results section the authors report that they “have/have not” detected differences using microscopy. First of all, it is not clear how they have quantified these differences.

Furthermore, they should state at what magnification the images were obtained.

According to the Reviewer's remark we have concluded the method section with an explanation of the confocal microscopy method and quantification. We have added the magnification scale on the images.

In addition, looking at the figures, it can be seen that cells have different morphologies. The authors did not comment on this. Can morphological changes be due to the presence of MTA? Please clarify this point and discuss these results.

Transformation of THP-1 monocytes into macrophages entails morphological changes - it's a normal feature of these cells. As was indicated in Figures, there were no morphological abnormalities in both types of cells due to incubation with MTA.

11) The authors should discuss the increases detected in their control cells.

There were no statistically significant changes in the controls 24h and 48h.

12) It would be interesting if the authors also included co-culture tests with monocytes/macrophages and an osteoblastic-like cell line to reproduce an in vitro model of a real-life condition.

We are grateful for this remark, however we are unable to include osteoblastic-like cell line in the manuscript. To do this, we would have to conduct another study on cells which, given the present health situation, is impossible due to limited access to the laboratory (the hospital is a Covid facility) as well as financial limitations to continue the studies. We sincerely hope that the Reviewer will take this into consideration.

13) Whole of discussion is poorly written and insufficient. The authors should improve the comments related to their results and should expanded the text with comparisons in details to previous literature.

Thank you for this remark, however we disagree with the opinion about the manuscript being poorly written and insufficient. The present study is indeed a pioneering study and it is difficult to refer to other results on MTA in the literature on the subject. Also, the model applied in the study is a novel approach, as has been noticed by other Reviewers.

14) It is not clear why the authors have included Figure 1. Figure 1 does not include data related to the experiments performed in this study. The authors should better motivate the presence of this Figure, otherwise they should remove it.

Thank you for this comment, though we would prefer that not only dentists but also scientists found out study interesting. Therefore, in our opinion, the picture showing a clinical use of the analysed material should be kept in the manuscript. Moroever, other Reviewers did not raise any concerns regarding this issue and complemented the clinical as well as scientifical merit of the paper.

Minor

  1. a) Figures are not cited in the text.

According to the Reviewer's remark we have cited Figures in the text.

  1. b) The authors improperly use the term “immunohistochemistry” in the text.

According to the Reviewer's remark we have corrected “immunohistochemistry” to “immunocytochemistry”

  1. c) The English language needs thorough revision by an English native speaker.

We enclose a certificate of a native speaker.

Round 2

Reviewer 3 Report

In the revised version of the manuscript the authors partially addressed all my concerns. Please find below a point by point response to the authors' comments.

1) I agree with the authors that a description of MTA is useful for a proper understanding of the issue. My suggestion was only to better summarize these points, leaving only the information essential for the understanding of the results presented in the study. For example, I believe that the description reported from line 403 to line 441 could be shortened or moved to supplementary materials.

2) Addressed.

3) Addressed.

4) Reading the paper, it is clear that cells were not used for further culture purposes. However, my concern was only related to the fact that during the freezing/thawing phase cells could be damaged. Since the aim of the authors was to evaluate proteins by different techniques, I was wondering why the authors had not stored the cell lysates or why they had not added proteases inhibitors before freezing.

5) I apologize if my request was not clear. My observation was not related to the order and structure of the sub-chapter 4.4.1. I think that the subchapter 4.4.1 is necessary. My point is that, going through the “Materials and methods” section, sometimes the authors tend to anticipate details that not pertain to the method they are describing in that point. This is not a problem if the reader reads the whole section, but it becomes a problem if the reader searches for information in a targeted way and does not find it in the right position in the manuscript. My intent was only to improve the manuscript by making it easier to consult for the readers of the journal. However, I take note of the author’s response.

6) Addressed.

7) I thank the authors for the response. At this point, I only ask the authors to substitute in all the manuscript “MTA” with “MTA HP Repair” to avoid misunderstanding.

8) Addressed.

9) Addressed.

10) I believe this point should be included in the text.

11) This is not what the authors have presented in their graphs. I think the authors should explain this point.

12) This point was only a reviewer’ curiosity and a suggestion for the authors. I can understand the difficulties due to this particular moment. I hope the authors will be able to be safely back in their labs as soon as possible.

13) The criticism I have raised was only related to the “Discussion” section and not to the whole manuscript. As I wrote at the beginning of my report, my opinion is that the topic of the work is interesting.

14) I don’t agree with the authors and I think that this should be included in the supplementary material at the most.

Minor

a) Addressed.

b) I believe that the correct term is “immunofluorescence”.

c) Addressed.

Author Response

Comments and Suggestions for Authors

In the revised version of the manuscript the authors partially addressed all my concerns. Please find below a point by point response to the authors' comments.

1) I agree with the authors that a description of MTA is useful for a proper understanding of the issue. My suggestion was only to better summarize these points, leaving only the information essential for the understanding of the results presented in the study. For example, I believe that the description reported from line 403 to line 441 could be shortened or moved to supplementary materials.

According to the Reviewer’s remark this part was shortened.

4.2. Chemical properties of MTA

Apart from portland cement (75%), bismuth oxide (20%), gypsum (5%), MTA patent specification lists calcium oxide (50-75 wt %) and silicon oxide (15-20 wt %) – 70-95% in total. Blending raw materials produces tricalcium silicate, dicalcium silicate, tricalcium aluminate, tetracalciumaluminoferrite [24]. The two available colours are due to Al2O3, MgO and FeO concentration [24]. The traditional composition of MTA showed long setting time, tooth and marginal gingiva discolouration, difficulty handling [2, 25]. The latter was reported at filling of root-end cavities, furcation and root perforation [1]. To counter this, MTA Repair HP (Angelus, Londrina, PR, Brazil) or MTA Vitalcem were developed [2]. MTA HP powder contains tricalcium silicate, dicalcium silicate, tricalcium aluminate, calcium oxide, calcium carbonate (filler material) and calcium tungstate (radiopacifier). The mixing liquid consists of water and plasticizing agent. It shows high-plasticity and improved physical properties [1]. Suggested applications are: root-end filling, pulp capping, pulpotomy, apexogenesis, apexification and repairing root canal perforations. MTA Vitalcem contains zirconium dioxide as radiopacifier and is suggested for root-end filling, perforation repair, root resorption, apexification and pulp capping. Its antimicrobial and regenerative properties are similar to the conventional MTA [26]. Table 2 presents detailed manufacturer data on the use of MTA.

2) Addressed.

Thank you

3) Addressed.

Thank you

4) Reading the paper, it is clear that cells were not used for further culture purposes. However, my concern was only related to the fact that during the freezing/thawing phase cells could be damaged. Since the aim of the authors was to evaluate proteins by different techniques, I was wondering why the authors had not stored the cell lysates or why they had not added proteases inhibitors before freezing.

We are very grateful for this remark. Of course, it is always a standard procedure to add a protease inhibitor to the frozen samples. Of course, it is always standard to add a protease inhibitor to the frozen samples.

Oczywiście standardem jest dodawanie inhibitora proteazy do zamrożonych próbek.

Of course, all the standard protease inhibitor is added to the frozen samples.

Oczywiście do zamrożonych próbek dodaje się cały standardowy inhibitor proteazy.

Nie mogę wczytać wszystkich wyników

Ponów próbę

Ponawianie próby

Ponawianie próby

We supplemented this oversight in the methods according to the Reviewer’s remark.

Next, cold PBS and protease inhibitor cocktail (Merck, Poznań, Poland) was added to cellular sediments and the samples were frozen at -80°C for further analysis.

5) I apologize if my request was not clear. My observation was not related to the order and structure of the sub-chapter 4.4.1. I think that the subchapter 4.4.1 is necessary. My point is that, going through the “Materials and methods” section, sometimes the authors tend to anticipate details that not pertain to the method they are describing in that point. This is not a problem if the reader reads the whole section, but it becomes a problem if the reader searches for information in a targeted way and does not find it in the right position in the manuscript. My intent was only to improve the manuscript by making it easier to consult for the readers of the journal. However, I take note of the author’s response.

We are very grateful for this remark, thank you.

6) Addressed.

Thank you.

7) I thank the authors for the response. At this point, I only ask the authors to substitute in all the manuscript “MTA” with “MTA HP Repair” to avoid misunderstanding.

Thanks again for the reviewer's attention. To make it easier to understand the nature of the diversity of MTA, it was explained in the discussion for the non-dentist reader that there are several types of MTA. This diversity results from the continuous improvement of materials in order to eliminate the drawbacks and increase the therapeutic benefits of this material. Unfortunately, in many works, the authors do not describe in detail which MTA material the description concerns, whether it is MRA Gray, white or HP. Wherever it was possible (ie wherever the authors of other works included it in their works) we made every effort to write the MTA type in detail. We found it superfluous to describe in detail the properties of each type of MTA, because the discussion includes a description of the properties of the inflammatory reaction (referring to the purpose of our work) related to the action of preparations under the common name of MTA. There are many other works (revue) on the types and characteristics of the physico-chemical properties of MTA, which the interested reader can refer to using our References. 

8) Addressed.

Thank you

9) Addressed.

Thank you

10) I believe this point should be included in the text.

According to the Reviewer's remark we have concluded the method section with an explanation of the confocal microscopy method and quantification.

In order to quantitative evaluation of enzyme expression cells were incubated with antibodies at the same concentration and condition of microscopic studies and washed three times with PBS at room temperature. Fluorescence intensity was measured in microplates dedicated for fluorescent studies (Eppendorf) by Asys UVM 340 microplate reader (Asys Hitech Gmbh, Austria) at the wavelength specified above. Fluorescence was normalized to protein levels measured by Micro BCA™ Protein Assay Kit (Thermo Scientific, Pierce Biotechnology, USA).

We have also supplemented the captions under the figures.

There were no differences in enzymes protein expression (p≤0.5, U Mann-Whitney test) examined by examined by measuring the intensity of fluorescence through a plate reader (data not shown). Scale bar 40μm.

11) This is not what the authors have presented in their graphs. I think the authors should explain this point.

Indeed, we apologize for the unintentionally incomplete answer. In our study we observed differences in the expression of MMPs between control 24 and control 48 groups. Such an increase in the expression of MMP-2 and MMP-9 in THP-monocytes and THP-1 macrophages was also observed by us in our earlier studies performed on this model. In our study, we observed differences in the expression of metalloproteinases between control 24 and control 48 groups. Such an increase in the expression of MMP-2 and MMP-9 in THP-monocytes and THP-1 macrophages was also observed by us in our earlier studies performed on this model.

W naszym badaniu zaobserwowaliśmy różnice w ekspresji metaloproteinaz między grupami kontrolnymi 24 i kontrolnymi 48. Taki wzrost ekspresji MMP-2 i MMP-9 w monocytach THP i makrofagach THP-1 zaobserwowaliśmy również we wcześniejszych badaniach przeprowadzonych na tym modelu.

In our study, we observed differences in the expression of MMPs examined between the control group and 24 control 48. Such an increase in the expression of MMP-2 and MMP-9 in THP-monocytes and THP-1 macrophages was observed by us also in our previous work done on this model.

W naszym badaniu zaobserwowaliśmy różnice w ekspresji MMP badanych między grupą kontrolną a 24 kontrolną 48. Taki wzrost ekspresji MMP-2 i MMP-9 w monocytach THP i makrofagach THP-1 zaobserwowaliśmy również w naszej poprzedniej pracy wykonanej na tym modelu.

Nie mogę wczytać wszystkich wyników

Ponów próbę

Ponawianie próby

Ponawianie próby

We think that such increase in these enzymes is due to the natural biology of these cells, so we have made separate appropriate controls for comparison with the appropriate study group.

12) This point was only a reviewer’ curiosity and a suggestion for the authors. I can understand the difficulties due to this particular moment. I hope the authors will be able to be safely back in their labs as soon as possible.

We are very grateful for this remark, thank you.

13) The criticism I have raised was only related to the “Discussion” section and not to the whole manuscript. As I wrote at the beginning of my report, my opinion is that the topic of the work is interesting.

We are very grateful for this remark, thank you.

14) I don’t agree with the authors and I think that this should be included in the supplementary material at the most.

Figure 1 is an essential element of the present paper, in the opinion of the authors. As their affiliations show, 3 of them are endodontists i.e. practitioners in the field of pulp disease treatment. The substantial value and merit of the clinical study presented in Fig.1 emphasises the practical and clinical applications of the obtained results of the laboratory studies. The 2 year-long positive clinical observations following the application of MTA-based formulation for covering the perforation of the bottom of the chamber, only confirm and substantiate the clinical value of the present paper, not to mention its importance in practical terms.

Moreover, the results of the laboratory assays are vital in terms of further practical applications as the use of the said material indeed is a commonplace practice of every endodontist. The clinical value of the presented results pertains to demonstration of the anti-inflammatory properties of MTA Repair HP, which was illustrated in Fig.1.

The confirmed anti-inflammatory properties of MTA Repair HP constitute a proof of its safety of use in clinical practice.

The experiments conducted in the course of the present study, confirm the manufacturers’ information on the biocompatibility and biointegration of the material used in the experiments with the cells of the host’s immune system. Therefore, we strongly insist that Fig.1 is a crucial element of the paper and should remain its integral part, rather than be moved to supplementary files which are rarely even considered by the readers.

 Minor

  1. a) Addressed.

Thank you

  1. b) I believe that the correct term is “immunofluorescence”.

We corrected term according to the Reviewer’s remark.

  1. c) Addressed.

Thank you
